# Bioinformatics for Marine Products: An Overview of Resources, Bottlenecks, and Perspectives

**DOI:** 10.3390/md17100576

**Published:** 2019-10-11

**Authors:** Luca Ambrosino, Michael Tangherlini, Chiara Colantuono, Alfonso Esposito, Mara Sangiovanni, Marco Miralto, Clementina Sansone, Maria Luisa Chiusano

**Affiliations:** 1Department of Research Infrastructures for Marine Biological Resources, Stazione Zoologica Anton Dohrn, Villa Comunale, 80121 Naples, Italy; luca.ambrosino@szn.it (L.A.); michael.tangherlini@szn.it (M.T.); chiara.colantuono@szn.it (C.C.); mara.sangiovanni@szn.it (M.S.); marco.miralto@szn.it (M.M.); 2Department of Cellular, Computational and Integrative Biology - CIBIO, University of Trento, 38123 Povo, Trento, Italy; alfonso.esposito@unitn.it; 3Department of Marine Biotechnology, Stazione Zoologica Anton Dohrn, Villa Comunale, 80121 Naples, Italy; clementina.sansone@szn.it; 4Department of Agriculture, University of Naples Federico II, Portici, 80055 Naples, Italy

**Keywords:** bioinformatics, omics, marine resources, biotechnological applications, marine observatories

## Abstract

The sea represents a major source of biodiversity. It exhibits many different ecosystems in a huge variety of environmental conditions where marine organisms have evolved with extensive diversification of structures and functions, making the marine environment a treasure trove of molecules with potential for biotechnological applications and innovation in many different areas. Rapid progress of the omics sciences has revealed novel opportunities to advance the knowledge of biological systems, paving the way for an unprecedented revolution in the field and expanding marine research from model organisms to an increasing number of marine species. Multi-level approaches based on molecular investigations at genomic, metagenomic, transcriptomic, metatranscriptomic, proteomic, and metabolomic levels are essential to discover marine resources and further explore key molecular processes involved in their production and action. As a consequence, omics approaches, accompanied by the associated bioinformatic resources and computational tools for molecular analyses and modeling, are boosting the rapid advancement of biotechnologies. In this review, we provide an overview of the most relevant bioinformatic resources and major approaches, highlighting perspectives and bottlenecks for an appropriate exploitation of these opportunities for biotechnology applications from marine resources.

## 1. Introduction

The origin of life has been traced from the sea about 1.5 billion years before the evolution of mankind. Since then, marine organisms have diversified in structure and functions, making the marine environment the largest and most variable ecosystem on Earth, comprising more than 70% of the planet surface and adapting to a wide range of conditions, from the extreme cold of polar seas to the extreme high temperatures and pressures of deep-sea hydrothermal vents [1].

The first living organisms appeared in the sea more than 3.5 billion years ago [2,3] and the evolutionary processes have molded marine organisms, which range from viruses to eukaryotes, to survive extreme temperatures, variable salinity and pressure, and attacks by other species, including prokaryotic and viral invaders [4,5,6,7,8,9,10,11,12,13].

The adaptation to a variety of conditions featured by extremely different marine environments determines an enormous amount of genetic and functional diversity [14], offering a precious source of biological materials and molecules which are contributing to innovation in many fields [1], including medicine and pharmacology [15], nutrition [16,17,18], agriculture [18,19,20,21], biofuels [22,23,24], cosmetics [25,26,27], innovations for sustainability (e.g., bioremediation [28] and bioplastics [29]), and other industrial sectors. As other examples, the marine microbiota appears to be a promising and endless source for new drug development [30,31,32], with new chemotherapeutants, novel antibiotics and health products to prevent and combat diseases [15], cancer [17,33,34], and drug-resistant pathogens [35], which are becoming a significant threat to public health. In health sciences, many marine natural products were revealed to be toxins or bioactive compounds, and were deeply studied to understand their action [15,17,33,36] and possible applications. In food sciences and agriculture, the marine environment has always been a gold mine [16,17,18,19,20,21], even when exploited as by-products or waste materials [19,20]. Marine products such as the algae-derived polysaccharides (e.g., agar), which have been used in food processing and preservation since the first half of the last century [21,37], are now widely used in nutrition but also for the delivery of bioactive compounds and nutraceuticals [38], or even for innovative opportunities (e.g., to produce degradable bioplastics [29]) for sustainable products.

Nevertheless, the marine habitat is still poorly explored. It is estimated that, despite 250 years of taxonomic classification and over 1.2 million species already catalogued in reference databases such as the World Register of Marine Species [39,40,41,42], 91% of species in the ocean still await description [43].

One of the reasons for the expanding interest in tools and approaches for observing and exploring the marine environment is to identify novel molecular entities as sources for new compounds for innovation in health, nutrition, agriculture, care, goods, and energetics.

Today, about 7000 molecules extracted from the sea are already used or are being validated for several purposes, ranging from medicine to industrial applications. The number of compounds isolated from marine species increases annually by almost 400 to 500 newly discovered products, and many more are still to be discovered [44]. Such compounds can be natively produced as secondary metabolites and become part of the organisms or secreted in the extracellular milieu [45]. Bioactive compounds can either be polypeptides or small molecules (lipopolysaccharides, polyphenols, alkaloids, etc.), but also nonribosomal peptides (e.g., vancomycin or daptomycin, actinomycin D, and cyclosporine) [46], polyketides [47], and nucleic acids [48,49,50]. However, the number of approved and marketed marine natural products is still very limited (11 approved drugs, five of which have anticancer activity, and more than 20 other natural products in clinical phase, as of 2018 [51]).

The trend to discovery novel products is focused mainly on the study of target species that are useful for the isolation of new active compounds, following well-established step-by-step approaches. Companies and enterprises are, therefore, strongly investing in all the methodologies that show potential to effectively shorten the pipelines for drug identification and development [52,53]. Research has focused on opportunities to generate more innovation in a short time and, as a consequence, is resulting in a remarkable interest in bioinformatics [54]. Indeed, bioinformatics offers methodologies for efficiently extracting value-added information from omics experimental data and for modeling, and these tools are useful not only to accelerate the identification of biologically active candidates, but also to investigate their action and effects on other living systems, such as on specific species or particular ecosystems (for example, in pharmacogenomics [55], microbial ecology, and agriculture [56,57]).

Since the capacity for producing bioactive compounds is encoded in the genome of a species, the identification of novel compounds and the molecular mechanisms for their synthesis often starts from genome or transcriptome sequencing, which takes advantage of the flourish of advanced methodologies which have been revolutionized by the advent of the next-generation sequencing (NGS) technologies, in the framework of the “isolate and then test” instead of “test and then isolate” approach [58]. Marine biotechnology, and biology in general, have largely profited in recent years from the advent of cost effective NGS, therefore, expanding the sequencing projects, which has resulted in major advances in the field [59], as well as other omics approaches (e.g., proteomics and metabolomics) [60,61], all supporting the understanding of structure and functionality of molecules. After the identification of novel compounds and possibly the elucidation of the metabolic pathways leading to these products, the organisms can be isolated and further investigated, or only the genes encoding the involved compounds can be isolated and then expressed in heterologous hosts, for controlled production. Alternatively, heterologous expression of biosynthetic genes or gene clusters (identified, for example, through metagenomics libraries) also produces compounds derived from yet-to-be isolated microorganisms [62] with cost effective techniques. On the one hand, isolated species, or the recombinant species, can be grown under controlled conditions (for example, in bioreactors) in order to obtain large amounts of the target compounds without harvesting the original wild population (which in some cases could lead to ecosystems unbalance) or using synthetic production that is often more expensive, thus improving sustainability of the productivity chain [25,63,64,65]. On the other hand, only a fraction of the marine diversity can be aptly cultured in a laboratory, and therefore this approach must be complemented by alternative techniques in order to explore a larger portion of this diversity. Indeed, in particular, molecular techniques and advanced sequencing approaches have been used for this purpose to capture the genetic and genomic diversity of the unculturable fraction of marine biological diversity (especially regarding prokaryotes).

The first massive exploration of marine diversity using a molecular approach resulted from the Craig Venter’s Global Ocean Sampling expedition. This scientific endeavor consisted of a worldwide voyage, inspired by Darwin’s voyage on the “Beagle”, undertaken to sample marine organisms and assess their diversity through DNA sequencing [66,67]. By analyzing mainly nucleic acid sequenced data and building in silico models, in fact, novel species were identified and the biosynthetic production of fluxes of compounds were inferred, understanding the definition of pathways of interest, and, eventually, redesigned [68]. For example, by means of advanced bioinformatic pipelines it is possible to identify thousands of possible biosynthetic gene clusters along DNA sequences, which can be explored and investigated by computational analyses before experimental characterization [69]. 

The ability to perform genome and gene data mining as an essential complementary approach to traditional experimental methods has significantly sped up the process of natural compounds discovery [70]. A closer look at the scientific production related to natural compounds obtained from marine species is found in the MarinLit database (Table 1). 

In addition, recent advances have strongly improved the biotechnological tools to enhance and manipulate the production of natural molecules. As an example, advanced experimental techniques such as those related to genome editing have been recently introduced and are being widely exploited to directly modify the genome sequence in regions of interest [71]. As a main novel example, approaches that take advantage of the CRISPR/Cas9 machinery [72] are being adopted to generate mutant genomes that target specific genes. Additionally, for these approaches, bioinformatics provides tools able to predict CRISPR/Cas9 targets, even in novel or partial genome sequences [73,74,75].

In this review, we focus on the main resources of bioinformatics and methodologies, discussing their role in supporting and accelerating the discovery of new marine-derived products, describing major applications, and highlighting opportunities, bottlenecks, challenges, and perspectives in the field.

## 2. Bioinformatics Applications and Resources in Marine Omics

A number of different approaches can be used to explore novel and useful compounds from marine resources, including metabolites, enzymes, or other molecules, and to investigate the molecular mechanisms involved in their production and functional properties. These methodologies range from whole-genome sequencing (a stand-alone research line which also provides a reference baseline for further omics approaches such as transcriptomics and proteomics used to investigate the functional activity of species or tissues) to metabolomics, used to understand the phenotypical effects of genome expression. Their meta-omics counterparts (e.g., metagenomics and metatranscriptomics) are able to tackle similar issues at the community level.

The following sections provide an overview of the main bioinformatic methodologies, with details on possible applications in marine biotechnology. A discussion of general topics that are not specific to marine biology, such as genomic or transcriptomic assembly and annotation, is beyond the scope of this paper, however interested readers can find pointers to relevant literature in the related sections.

### 2.1. Genomics and Transcriptomics

The advent of novel technologies, such as the introduction of next generation sequencing (NGS) techniques, favored the shift from the less efficient Sanger methodology to the sequencing of huge numbers of DNA fragments, due to fast and cheaper high-throughput technologies. The BAC-by-BAC (Bacterial Artificial Chromosome) based genome sequencing was almost replaced by the whole genome shotgun (WGS) approach [76]. This transition increased the need for new methods of data processing, mining, and management and further challenged bioinformatic research to provide advanced technologies to support the sequencing efforts [77]. This change resulted in the establishment of several genome-sequencing projects, expanding the activities that were mainly focused on reference model species for marine biology, such as *Ciona robusta* or *Strongylocentrotus purpuratus* [78,79], to other species, with the release of many draft genomes obtained from the sequencing of new species and the resequencing and genotyping of already available genomes [80,81,82,83,84] (Figure 1). Resequencing efforts were, in some cases, necessary to improve the poor quality genomes obtained with older or inadequate technologies, which were useful to detect candidate new compounds but not for comparative genomic analyses. 

Under the umbrella of the International Nucleotide Sequence Database Collaboration (INSDC) [85], all the information related to biological sequences, including those from marine resources, is flowing into general databases (Table 1). The Reference sequence database at NCBI [86,87,88]; the EMBL-EBI sequence collection, including the vertebrates, prokaryotes, protists, fungi, plants, and metazoan partitions [89,90]; and the DNA Data Bank of Japan (DDBJ) [91] are the three reference sites in the consortium. In addition, due to the generation and release of huge numbers of sequences (raw reads) produced and released by next-generation sequencing efforts, the INSDC system built specialized archives to store data either as raw or processed, such as the Sequence Read Archive (SRA) [92] and Gene Expression Omnibus (GEO) [93] at NCBI, ArrayExpress [94] and European Nucleotide Archive (ENA) [95] at EMBL-EBI, and the DDBJ sequence Read Archive (DRA) [96].

Beyond the INSDC project, the Integrated Microbial Genomes with Microbiome Samples (IMG/M) and Expert Review (IMG/ER) [97] at the Joint Genome Institute (JGI) and proGenomes [98] are parallel efforts organizing reference resources for microbial genomes and microbiomes, providing information on genes, genomes, and functions, and providing tools for comparative analyses. In more detail, the IMG/M and IMG/ER partitions at the JGI provide highly specialized repositories for curated microbial, viral, and fungal genomes with taxonomic affiliation and specific tools for exploring their characteristics (e.g., assembly quality and completion levels, potential markers for auxotrophy, and geographic localization). The proGenomes initiative represents an attempt to provide a highly accurate prokaryotic genome database with curated taxonomic affiliations and functional annotations based on different collections, including CAZymes and dbCAN [99,100], and markers for antibiotic resistances, which represent useful references for the selection of organisms of biotechnological interest.

The Kyoto Encyclopedia of Genes and Genomes (KEGG) genome partition within the KEGG resource [101] also acts as a reference repository for sequence data, which can be queried by users to characterize enzyme pathways and explore potential genes of biotechnological interest in complete reference genomes.

In order to annotate genomes, several resources are available to provide functional information about gene and gene clusters, beyond the gene association to pathways as provided by KEGG. One example is the Gene Ontology (GO) international consortium, which aims to provide reliable gene classification based on their functional descriptions and on the establishment of a reference vocabulary of molecular functions, cellular locations, and biological processes that gene products may be involved in [102]. As an example, one of the most used platforms for searching and browsing the Gene Ontology database is represented by AmiGO [103]. A widespread use of the GO is to perform gene sets enrichment analysis. Given a set of genes, for example those that may be expressed in specific conditions, an enrichment analysis can detect the over- or under-represented GO terms within the selected dataset as compared to a species-specific GO collection for the whole gene complement [102]. However, even using different tools to functionally annotate genomes, many genes still remain undefined. The percentage of anonymous genes can be very different among different species or taxa.

Additionally, the transcriptome sequencing shifted from the low-throughput Sanger-based expressed sequence tags (ESTs) production [104] to NGS approaches. Among these, RNA-seq produces a more detailed and quantitative overview of a transcriptome and the associated level of expression per gene, also providing—thanks to dedicated bioinformatics pipelines [105,106,107]—deep details on alternative splicing and allele-specific information [108], even in the absence of a reference sequenced genome (de novo transcriptome analyses). Compared to other transcriptome-based approaches such as ESTs and microarray analyses [109], the throughput of the RNA-seq techniques, together with lower experimental costs, allowed the spread of many projects that either accompanied the genome sequencing of many species to define their representative gene expression atlases [110] or independently allowed characterization of the transcriptome complement of a novel species exploiting a de novo assembly approach [111,112,113,114,115,116,117,118,119,120].

Nucleic acid sequencing techniques are also used a great deal in marine biotechnology and, in particular, to search for new marine drugs, due to the combination and integration of genomics and transcriptomic approaches that aim to find and quickly annotate genes producing interesting compounds [121,122]. Moreover, genomics and transcriptomics have been proven to be useful for the characterization of marine species that are important in the production of secondary metabolites and enzymes of interest for industrial, pharmaceutical, and green biotechnology applications [123,124]. Some recent examples of such enzymes include the new flavin-dependent halogenase, isolated from a marine sponge metagenome [125] and several α-amylases isolated from a sea anemone microbial community [126], whereas metabolites range from derivatives of amino acids and nucleosides, macrolides, porphyrins, terpenoids to aliphatic cyclic peroxides, and sterols [127].

Transcriptomics can also help determine whether biosynthetic gene clusters are transcriptionally silent or not [128], by revealing their regulatory machinery, and possibly, the type of post-translational modification that can be amended to the proteins [129]. To this aim, we also mention some of the alternative NGS-based sequencing approaches that are being increasingly used to better support these investigations, such as small RNA, epigenome, or single cell sequencing [130,131,132,133]. We do not include specific details for associated repositories, although all the associated public sequencing production is collected in the general NGS-related resources previously mentioned in this review. 

The data collections from transcriptomic projects are also all available through reference databases. In particular, ESTs libraries are stored and easily retrievable in the dbEST partition of NCBI [104] (which was included in the nucleotide section since the beginning of 2019), while raw RNA-seq data are included in SRA [92], ArrayExpress [94], and DRA databases [96]. 

In some cases, it is also possible to exploit both genomic and transcriptomic data through dedicated web pages that can be species-, genera-, or clade-specific. For examples of marine species, remarkable marine-specific multi-omics resources are Aniseed [134], fully dedicated to sea squirts (Ascidiacea), and Echinobase [135], which include genomes and transcriptome data of five different echinoderms, and the genome projects of the OIST Marine Genomics Unit (Table 1), which includes information concerning 19 different marine species. All these resources allow the user to access both annotated genome assemblies, as well as raw reads, and are accompanied by genome sequence browsers [89,134] to visualize structures of genes and transcripts, and, when available, to retrieve information on the encoded proteins [136].

Because sequence-like data are the major reference product in molecular biology, the development of bioinformatic methodologies has focused extensively on the design of techniques to detect sequence similarities. Most computational methods for sequence similarities are based on global or local similarity searches that are based on alignment tools [137,138]. Currently, the methodology used most to detect similarities at the nucleotide or amino acid level is the basic local alignment search tool (BLAST) [139], which compares nucleotide or protein sequences to sequence databases. Since some particular BLAST searches can be very sophisticated and involve intense computations, such as tBLASTx analyses of entire transcriptome collections that consider all six potential ORFs of each sequence, similar efforts spread. Complete genome alignments can be carried out using different approaches, from nucleotide alignments (e.g., through the LAST sequence alignment toolkit, which also can be utilized for alignment of very large mammalian genomes [140]) to block alignments (e.g., through Mauve [141]). The genome alignment tools can highlight conserved regions, rearrangements, and differences between large genome sequences, allowing the researchers to analyze peculiarities at the genomic level (e.g., sequence insertions, duplications, or potential horizontal gene transfer events). This information can be used to search for potential novelty genes or operons, reorganization, and regions encoding for the production of novel molecules of biotechnological interest. 

Similarity search methods are also the basis of the approaches that focus on the definition of computationally based homologs, comparing genes or genomes based on orthology inference [142,143,144,145,146], analyzing gene families mainly based on the detection of computationally defined paralogs [143,147,148], and highlighting peculiarities due to the selection of those genes that are species-specific [149]. 

The aforementioned approaches were also adapted to detect highly related conserved portions of genomes, even in the same species. This is generally common in prokaryotes, in which genome plasticity, mosaicism, and high rates of horizontal gene transfer drove strain differentiation [150], although present also in more complex species such as plants and vertebrate that show variable levels of genome duplications [151,152]. This deluge of genomes belonging to the same taxon led to the development of the concept of pan-genomics [153], which refers to “the entire genomic repertoire of a given phylogenetic clade, encoding for all possible lifestyles carried out by its organisms.” To this extent, several different pipelines that perform these analyses have been developed over time. Some of them are available as tools for local installations (e.g., micropan for R [154] or PanFP [155]), others are instead available as webservices (e.g., PGAweb [156]). Tentative attempts at defining public databases to explore microbial pangenomes have also been devised (e.g., PanGeneHome [157]).

A comparison among several genomes of the same taxon helps researchers determine to which extent the actual genetic diversity has been sampled. For example, species such as *Bacillus anthracis* have a closed pangenome, with 2893 core genes and only 85 accessory genes after just nine individuals sequenced [158]. Conversely, species such as *Pseudomonas aeruginosa* have a relatively small core genome as compared with the large accessory genome (665 genes constituting the 1% of the whole pan-genome), and therefore its pangenome is defined as “open”, meaning that its diversity is still not sampled thoroughly [159]. From a biotechnological point of view, pangenomic analyses can highlight whether genes or gene clusters of biotechnological interest can be found in specific strains of well-known organisms, and thus potentially introduced in novel screening approaches or easily transferred to well-characterized organisms for their high-throughput production [160,161]. Indeed, this approach has been applied before to identify, clone, and express candidate antibiotic resistance genes in the *Salinispora* genus by screening more than 80 strains. This approach also correctly identified previously “orphaned” gene clusters, for which function could not be assigned, by inspecting the function of their orthologs and by analyzing their products through heterologous expression in suitable hosts [162].

Due to the diffusion of NGS technologies and the constant decrease in the sequencing costs a plethora of genomics and transcriptomic datasets have been generated. Although associated with the same species, they often exhibit high dissimilarities in terms of data quality, curation, and methodologies employed (e.g., same species with different genome annotation versions). This is due to several factors which include the following: (1) data production is several order of magnitude faster than the release of exhaustively annotated and curated datasets; (2) the opportunity to publicly release even partial and still uncomplete data [163], which can be of interest for the scientific research, but often remain in a preliminary version; (3) the run for releasing dedicated resources for specific targets, which often causes uncoordinated parallel efforts, resulting in similar resources covering partially overlapping information; (4) the lack of rules for the withdrawal of obsolete public collections; and (5) the presence of software errors, such as in automatic annotation pipelines, that might generate errors not easily detected and that are, if not curated, inherited in subsequent versions. Navigating this overwhelming amount of resources can cause confusion for non-expert users and lead to limited scientific applications, and thus determining one of the major bottlenecks in bioinformatics [163]. Examples of heterogeneity in terms of data content are evident even in reference platforms such as NCBI and Ensembl [87,89]. As an example, for the echinoderms reference species, i.e., *Strongylocentrotus purpuratus* (purple sea urchin), the latest genome assembly versions in NCBI and in Ensembl are different (Spur. version 4.2 and 3.1, respectively), misleading the users and affecting the reproducibility and the comparability of the generated results outside of the used platform. Moreover, as in the case of the diatom *Phaeodactylum tricornutum*, although the latest available genome assembly version is the same in both NCBI and Ensembl, the genome annotation versions refer to different analytical pipelines, the NCBI genome annotation pipeline [164], and the Ensembl gene annotation approach [165], respectively. Indeed, annotation pipelines have different sensitivity for determining gene structures and predicting CDSs, giving results that are resource dependent and do not necessarily fully overlap.

The sources of heterogeneity, in terms of genome assemblies and gene annotation versions, and the quality of the annotation, are severe limiting factors for the sharing of comparable results from public web-based services, which affect the reliability of the available information resources and subsequent results such as gene expression, gene family analysis, and comparative genomics. These issues are worsened when considering prokaryotic genome annotation. Thousands of prokaryotic genomes are released yearly and annotated with automatic tools, whose accuracy did not improve accordingly [166]. Actually, the sources of biases are amplified as an effect of draft annotations and a solution to this problem has not been found yet [166]. 

All these aspects may also mislead non-expert users, who need education in the field in order to appropriately move through the overwhelming amount of resources. To mitigate this issue, the straightforward direction should be that reference websites (e.g., NBCI and Ensembl) should share, cross-reference, and integrate more information, even coming from smaller consortia efforts, and clearly report updates and errors, if any. A complementary approach could be provided by experts in the field that could produce smaller but effective resources by releasing manually curated information on selected species. As an example, the GENOMA platform [167], an ongoing project, at the moment collects and integrates genomic information about four different marine species, reporting statistics and comparisons among several genomics resources, also through dedicated genome browsers.

### 2.2. Metagenomics and Metatranscriptomics

To date, it has been established that a very minimal fraction of all the microorganisms inhabiting marine environments can be successfully isolated through traditional culturing methods, and that the vast majority of the potential functional diversity in the ocean is currently not exploited [168,169]. However, the emergence of culture-independent techniques coupled with metagenomic approaches has provided researchers with additional and valuable tools to analyze the functional potential of a community of species [168,169], which is a key approach to detect novel opportunities for biotechnological applications. 

Metagenomics is a widely explored approach, promoting major shifts in understanding marine ecology. Specifically, metagenomics refers to the genetic and genomic analysis of microorganisms recovered from mixed communities from a specific environment [170] and can be utilized for the taxonomic and functional characterization of that environment. One example was the identification of proteorhodopsin, which led to the discovery of new trophic strategies in the ocean surfaces [171,172]. 

The advent of massive DNA and RNA sequencing technologies has enabled the development of large-scale research endeavors. After Craig Venter’s seminal Global Ocean Sampling expedition [66,67], many others have been carried out, such as the Tara Oceans and Malaspina expeditions [173], which are leading marine researchers that explore novel tools to appropriately study this huge diversity. Although promising, these approaches are inherently complex. Depth of sequencing, library construction technique, and sequencing technology, for instance, can either enrich for specific fractions of the marine community or generate biases [174,175,176]. In addition, the bioinformatic approaches used in these studies are still evolving, constantly being improved to adapt to the challenges imposed by the complexity of this research and to the updating of reference information which is accumulating quickly. To support the validation of the best practices for achieving reproducible and reliable results, a comprehensive evaluation of such methods is being carried out in the framework of the initiative for the Critical Assessment of Metagenome Interpretation (CAMI) [177].

Recent investigations of the metabolic potential of genomes reconstructed from metagenomes defined thousands of genomes or their fragments from several marine environments [178,179,180]. Indeed, genome mining can be helpful for the identification of new compounds [181], whereas comparative genomics may lead to the inference of ecological or evolutionary patterns [182]. As an example, such approaches revealed an interesting functional partitioning between surface and deep-ocean populations of the clade SAR11 (which is one of the most abundant components of bacterioplankton) [183], which had important consequence on the global nitrogen balance [184].

As both an alternative and a complement to metagenomics, metatranscriptomics has recently been expanded and better explored to characterize complex natural communities from a functional point of view [185]. This approach, which has been used to characterized different ecosystems, including marine deep-sea sediments [186], provides advantages as compared to the DNA-based sequencing, which include minor susceptibility to amplification biases and the possibility to only capture the living fraction of the organisms inhabiting the community. Nevertheless, it is also characterized by important hindrances and potential biases, including but not limited to the lack of reference genomes (which are needed for the evaluation of potential taxonomical and genomic novelties) and standardized laboratory procedures and bioinformatics pipelines [187]. Therefore, there is a compelling need for the development of standardized reference collections and protocols for metatranscriptomic annotations and analyses, which are still quite pioneering but might yield important results for bioprospecting [185].

The massive flow of meta-omics sequence data highlights the need for comprehensive databases to collect the accumulating information and, additionally, appropriate curation and tools to exploit this information for taxonomical assignments and functional analyses. Until now, meta-omics-specific reference databases did not exist. However, efforts have been implemented for the creation of sequence databases which could act as both repository and data analysis reference sites, most providing either access to specific sequence databases or to more generalized repositories. One example is represented by the MGnify tool within the EBI metagenomics portal [188]. This tool makes it possible for users to characterize raw sequence data and assembled contigs using either a taxonomical (through analyses of sequences related to small and large ribosomal subunits) or a functional-based approach (through gene finding and analysis of potential protein coding nucleotide sequences as compared with data from the InterPro database and GO [189,190], see also the proteomics section for more details), thus utilizing publicly-available reference databases. Another reference example to the scope is provided by the MG-RAST server [191], which is a storage and analysis dedicated web server that allows the processing and storage of raw sequence metagenome data (Table 1). Established in 2008, MG-RAST currently stores up to 203.43 Tbp of 385,064 metagenomes [192]. It allows users to search for taxonomic and functional annotations of the submitted sequence samples using a combination of sequence databases. RefSeq for taxonomic annotation of shotgun reads and contigs, the SEED Subsystem architecture, KO, NOG, and COG databases for functional annotation and the RDP, GreenGenes, and SILVA databases for ribosomal subunit similarity.

More multipurpose databases and repositories are represented by the KEGG MGENES partition [101], which is an attempt to store and organize a collection of genes reconstructed from metagenomes, allowing the search for specific genes, the browsing of gene annotations, the comparison among samples, and the BLAST-based comparisons against the database, and by the MMP (marine metagenomics portal), a specialized repository for (meta)genomic data for marine microbial organisms [193]. Currently, this system provides the MAR databases (contextual and sequence databases of complete and draft marine prokaryotic genomes, as well as genes and proteins from metagenomic samples, which can be downloaded to be deployed locally for other purposes), the META-PIPE pipeline [194] (a workflow for the analysis of metagenomics data, not yet available to the general public) and MAR BLAST (a basic BLAST search tool against the MAR databases). This repository can provide useful information concerning taxonomic and functional data of marine prokaryotes, which can be further investigated and tailored to gain insights on species and genes of biotechnological interest.

More recently, the two expeditions Tara Oceans [195] and bioGEOTRACES [196,197,198,199] started collecting marine water samples with data generated from both projects, which were organized in different, independent databases. In particular, the Tara Ocean expedition gave rise to more general bioinformatic resources, as well as to specialized sequence repositories. As an example, the Ocean Gene Atlas [200], a web server organizing a collection of 40 million prokaryotic genes and greater than 110 million eukaryotic transcripts, which have been produced in the framework of the Tara Ocean investigations, allows for the query and comparison of nucleotide and amino acid sequences against the built-in databases. This server has been used, for instance, to investigate a new class of potentially widely distributed subclass of carbonic anhydrase which might play important roles in the global carbon cycle [201]. The GLOSSary (GLobal Ocean 16S subunit web accessible resource) [202] represents an effort to appropriately organize reprocessed taxonomic data from prokaryotes extracted from published Tara Oceans sequence datasets. The platform allows users to explore and query the underlying dataset to obtain indications on the distribution of prokaryotic organisms across the major oceanic basins. Although this specific platform currently exclusively relies on ribosomal data to investigate the taxonomic information within the Tara Ocean sequence data, it allows researchers to analyze the geospatial distribution of species of interest and also to gain insights into potential relationships, beyond providing data to complement experimental efforts and link genomic resources (including genes and gene clusters of biotechnological interest) to specific environments. 

As addressed by these examples, the reported attempts to organize (meta)genomic data are disparate and heterogeneous, and none of them are specifically focused on bioprospection and biotechnological developments, even though separate tools are being made available to this aim, such as, for example, the dbCAN meta-server [203], which was designed for the investigation of carbohydrate-active enzymes. Possible approaches for bioprospection and identification of novel useful compounds in metagenomics include the identification of genes or gene clusters for the discovery of secondary metabolites and catalysts for their synthesis [204] or novel enzymes at the whole-community level, or in silico isolation and characterization of genomes or associated portions (“genome-resolved metagenomics”). After screening through bioinformatic pipelines, mainly based on similarity searches, potential genes and clusters can be identified, cloned, and expressed in heterologous hosts [62,205], as previously introduced.

### 2.3. Proteomics and Structural Biology

Proteins are the main actors in functional processes carried out by a biological system. They act in response to the development of internal or external stimuli, and to environmental changes [206,207]. Proteomics aims to identify and quantify proteins, a systems-based perspective of how organisms mount their molecular responses.

The two-dimensional gel electrophoresis (2-DE) technique, which separates mixtures of proteins based on their properties, enables the dissemination of different proteomic approaches [208]. Bottom-up proteomics procedures include the proteolysis of protein mixtures, and the analysis of the generated fragments by liquid chromatography–mass spectrometry (LC-MS) [209,210,211]. In top-down procedures, proteins are directly subjected to gas-phase fragmentation, followed by MS analysis [212,213]. Middle-down proteomic approaches [214], instead, generate longer peptide fragments as compared to bottom-up strategies that use protocols involving single-residue specific proteases such as Lys-C [215,216,217], Glu-C [218,219], Asp-N [220], and Lys-N [221,222].

Proteomic studies led to the discovery of peptides and toxins useful for biomedical research from sea anemone [223], sea sponge [224], cone snails [225], and cyanobacteria [30,226]. In a focused special issue of 2015, entitled “Proteomics in marine organisms” [227], 20 contributions about different species ranging from Bacteria and mammals, to microalgae and flowering plants, provided a representative compendium on marine proteomics.

All proteomics studies, as all omics approaches, rely on bioinformatic resources that enable the analysis of the raw data, as well as the exploitation of the produced outcomes. The reference resource of protein sequences and their annotation is UniProt, The Universal Protein Resource [228], collecting more than 16,000 reference proteomes (updated to July 2019). This general resource offers a BLAST server to sequence similarities detection by scanning the entire UniProt database, a multiple alignment tool based on the Clustal Omega program [229], and a text search by keywords.

One of the most comprehensive resources in terms of information related to protein sequences is InterPro [190], a database containing different kinds of classifications of protein-related features, including, as an example, protein family information from PFAM, the protein families database [230], accompanied by further detailed descriptions such as protein domains or sequence conserved signatures. Users can perform a similarity-based functional annotation, and also list all the proteins across all the species in the InterPro database having the same functional annotations. InterPro developers, moreover, freely distribute an associated software to enable the users to retrieve information about thousands of protein sequences in one analysis [231].

An important branch of proteomics for biotechnological applications is the so-called structural biology [232]. Structural studies on protein data are important to understand which and how amino acid sequences contribute to a specific protein folding, revealing structure–function relationships, a fundamental step for the elucidation of cellular processes. Protein structure information, as examples, is essential to address challenges in enzyme discovery and to identify ligand receptor properties, favoring protein design. The prediction of three-dimensional (3D) structures, the investigation of structural peculiarities, the simulation of functional and structural behavior of biomolecules, as well as their interactions provide valuable predictive tools as an alternative to expensive screening experiments, which are crucial to the search for lead compounds in biotechnological applications, drug discovery, and design. The main bioinformatic applications downstream of structural proteomics techniques, indeed, are in the fields of (1) prediction of protein structures, (2) molecular dynamics simulation, and (3) molecular docking.

Prediction of protein structures is a fundamental approach to highlight conformational aspects of molecules of biotechnological interest, for example, elucidating structural features related to environmental adaptation, such as warm or cold-adapted mechanisms that confer thermostability in extremophilic enzymes [233,234], or specifying enzymatic action useful for biotechnological applications [235]. The prediction of protein structures follows two main strategies: (1) comparative approaches, based on homology modeling [236] or protein threading techniques [237,238], which predict new structures by modeling sequences from unknown structures using solved structures from homolog sequences as templates, or by recognizing common protein folds in protein sequences that lack homolog sequences; and (2) ab initio approaches [239], based on intrinsic chemical and physical characteristics of amino acid sequences rather than previously solved structures. Major protein prediction programs and web resources are summarized in the related section in Table 1.

Molecular dynamics simulations enable evaluation of the biotechnological potential of molecules of interest by providing an in silico estimation of the stability of enzymes and protein complexes even before performing in vitro studies [240]. Widely used software packages for performing molecular dynamics simulations are GROMACS [241] and NAMD [242], which simulate the Newtonian equations of motion for biological systems with hundreds to millions of particles. Other widely used molecular dynamics simulation programs [243] are listed in the related section in Table 1.

Molecular docking is an in silico drug design approach to leverage 3D structures for ligand discovery, fitting one or more compounds into binding sites [244,245], predicting the bound conformations and the binding affinity. AutoDock [246] is one of the most-used programs for molecular docking and virtual screening, particularly after the speed up deriving from the implementation of multithreading in the AutoDock Vina [247] update. SwissDock [248] is a public webserver based on EADock DSS software [249] and on S3DB—a database of manually curated target and ligand structures [250] that is able to predict complexes between proteins and small ligands. These bioinformatics approaches, together with cutting-edge technologies able to highlight physical interaction with the target protein (e.g., the cellular thermal shift assay (CETSA) [251]), are essential in the design and development of new drugs. Other molecular docking programs [252] are listed in the related section in Table 1.

A fundamental reference resource for structural bioinformatics applications is the Protein Data Bank (PDB), a database of tridimensional structure data. It stores structures from X-ray crystallography, nuclear magnetic resonance (NMR), cryo-electron microscopy, and theoretical modeling [253]. The expansion of these useful collections thanks to novel technologies in high-throughput structure determination is also going to provide a consistent boost to the current information [254].

The major bottleneck of proteomics studies derives from their nature, that is, from the complexity of biological structures and of the physiological processes in which they are involved [255,256]. In particular, in structural biology applications, this complexity has an impact on the resolution of the protein structure data generated by crystallographic or NMR experiments [257], affecting all the downstream bioinformatics procedures described above, such as homology modelling, molecular dynamics and drug design techniques. Structures with resolution of 3 Å or higher show only the basic conformation of the protein chain, lacking any information about their atomic structure [253]. Moreover, the need to isolate and study molecules through structure outside from their natural functional context, inhibiting their typical changes, can affect the right conformational assignments and mislead associated investigations on their behavior in biological environments.

### 2.4. Metabolomics

Metabolomics, together with other omics sciences, provides large and complex datasets, fundamental to understanding a wide variety of cellular processes. From this perspective, the extreme variability of chemical and physical conditions in the marine environment have made metabolomics a key field for the study of marine diversity [127]. The metabolome of an organism, in fact, directly correlates with gene expression and the associated protein production, affecting downstream functional pathways and representing the phenotypical responses of the organism to a vast range of physiological and environmental stimuli [258]. Often, organism reactions to a changing condition include the remodeling of their metabolism and regulating the levels of specific metabolites, which can potentially represent markers of a particular response (e.g., biotic or abiotic stresses). In this context, metabolomics helps the evaluation of the impact of climate changes on marine organisms, unraveling contributions that marine systems could play in mitigating the effects of global warming [259,260].

While MS-based proteomic approaches still require the separation of protein mixtures and the analysis of fragmented peptides, metabolomic approaches are based on the direct profiling of nonfragmented molecules via MS techniques [261,262]. 

Although bottlenecks make these technologies spread and improve their throughput, studies from metabolomics has helped, for example, to identify compounds with inhibitory effects against common human pathogens from the sponge bacterium *Rhodococcus* sp. UA13 [263], and from a panel of marine myxobacteria [264]. Other studies have suggested the use of marine-adapted fungi as biocontrol agents in agriculture [265]. 

Bioinformatics remarkable resources exploited in such efforts are the global natural products social molecular networking (GNPS) [266], which represents an open-access knowledge basis for organization and sharing of raw, processed, or identified tandem mass (MS/MS) spectrometry data, and the antiSMASH server [70], which allows genome-wide identification, annotation, and analysis of gene clusters related to secondary metabolite biosynthesis.

KEGG [101], Reactome [267], and MetaCyc [268] are the reference databases for enzymes, reactions, and metabolic and regulatory pathways, respectively. These resources include tools to highlight and interact with specific sub-paths or enzymes within the maps of the metabolic pathways of the selected species, enabling the download of maps in different file formats. An innovative software implemented in MetaCyc is the Pathway Tools software [269], which permits the computational prediction of the metabolic networks of any organism that has a sequenced and annotated genome [270].

ChemSpider [271] and The Super Natural II database [272] are two public resources providing access to the structure information of a huge diversity of compounds, including element composition, molecular weight, monoisotopic mass, and pharmacological activity.

NaPDoS [273] and MEROPS [274] are specific resources exclusively dedicated to secondary metabolite genes associated with polyketide synthase and non-ribosomal peptide synthesis pathways, and to peptidases, their substrates, and inhibitors. 

All the presented resources rely on the correct definition of the biological function of the collected bioactive compounds and metabolites. An accurate annotation is necessary for data interpretation; however, metabolite identification is still a major bottleneck in untargeted metabolomics [275]. Computational workflows for metabolomic interpretation, including high-throughput metabolite profiling and annotation, are highly challenging tasks, with fast evolving metabolomics datasets specifically generated by dedicated service centers [274,275]. The main delicate issues are due to the variability of resolution and the difficulty to establish generalized standards from different specialized laboratories and technologies. Although community guidelines for the detection of metabolites were established years ago, the adaptation to recommended standards is still far from being achieved. The complexity of metabolomic data from different combinations of various chromatographic and mass spectrometric acquisition methods has resulted in the establishment of diverse pipelines and workflows, which often involve nonstandardized manual curation. Furthermore, bioinformatics tools in the field still need to better address the problem of enrichment analyses, accurately linking metabolomic data to the most reliable similar compounds and building exhaustive pathway diagrams. These approaches need to be integrated into customizable workflows, such as the ones based on R or Python programming languages for the design of reusable and shared software [276].

The future of metabolite identification depends on the use of metabolome data repositories and associated data analysis tools, enabling data sharing and downstream analyses in an automated fashion, overcoming the lack of standardized methods or procedures [277,278,279].

## 3. Bottlenecks and Perspectives

### 3.1. Bottlenecks

It is evident from the many examples given above that marine organisms represent an ever-increasing subject of scientific investigation. Of course, marine organisms are also of great commercial interest for big (chemical, pharmacological, biotech) companies, especially those species belonging to the so-called marine areas beyond national jurisdiction (ABNJ), which cover 64 percent of the surface of the oceans and nearly 95 percent of its volume. Usually rich entities, such as big companies or first-world universities, which have invested money in the collection and analysis of marine organisms, secure their findings by national or international patents [280], thus denying the scientific community free access to their results. Although the Nagoya protocol (2010) has somehow addressed the need for regulation in the field, the question of who is the owner of the ocean’s biodiversity data still remains open, and what kind of legal and political action are needed to prevent an unfair appropriation of marine data is still a subject of debate [280].

The establishment of general centralized data repositories through reference sites and exploitable through methodologies publicly accessible due to extended bioinformatic efforts in the different fields of the omics technologies, was an incredible achievement in biosciences, favoring the sharing and the spreading of information fundamental for the fast advancement of the scientific research. These efforts reduced the scientific costs, due to the redistributions of methods and results and paved the way for further investigations offering general benefits for all scientific communities. On the one hand, flourishing community-specific collections and the accessibility to these approaches, even for non-expert users, need a conscious set up also of shared rules and appropriate education to avoid spreading limited quality and poorly reusable datasets. On the other hand, prior to any bioinformatic approach, correct taxonomic information on the specimens used is paramount for the production of high-quality research: sadly, the number of expert taxonomists in many fields of biologic research is dwindling, in some cases due to the difficulties taxonomists face in the identification of specimens. In addition, specialized backup facilities are also required to maintain voucher collections for future inquiries.

Although, on one hand, the fast evolving performances of sequencing technologies are the core of the incredible acceleration of molecular data production at affordable costs, on the other hand the fast production and release of novel sequence data, such as those from genome or transcriptome assemblies, faces the bottleneck of slower bioinformatic research to establish curated collections and updated information, even in reference platforms. The constant fast release of novel sequenced genomes or different assemblies from the same genome, as an example, reduces the efficiency of data curation, resource updating, and, as a consequence, may affect the quality of the subsequent analyses, such as gene family or gene expression assessments, as well as comparative genomics [144]. This holds for all the sections described in this review. Instead, specific bottlenecks that are exclusively related to a particular area of omics sciences, are discussed within each section of this review.

Bioinformatic tools need to follow the fast development of novel technologies and to adapt to the larger data size, but there is also a need for expert curators and of coordinated community feedback to validate the spreading results. Indeed, limits in data annotation and its updating and curation represent the main challenges in this field of research in biology.

### 3.2. Perspectives

Because of the precious amount of information that the sea can offer, the establishment of integrated marine data collections from multiple observations is emerging as a compelling need in the scientific community, primarily with the aim of assessing and monitoring the health status of marine ecosystems, but also as a multifaceted approach to unravel the complexity of marine species of biological and biotechnological interest [281,282]. These marine observatories are expected to support the collection, in both time and space, of several different types of information, ranging from classical biogeochemical and oceanographic measurements, to satellite images and multi-omics molecular data.

The enormous flow of data generated by the heterogeneous technologies which are being employed could initially appear as overwhelming when focusing on comparisons between different datasets. Big data should always suggest focusing on data acquisition, collection, and organization rather than data quality assessment as a first step. Data should always be considered to be an important factor of richness and one of the major opportunities for advancement, as long as efforts are made in the direction of their collection, integration, standardization, interrogation, and interpretation. In the past, the budget requirements for storing data have been quite disadvantageous; however, during the last decade, storage costs have become cheaper, due to hardware technologies and cloud commercial offerings, making a catch-all approach possible or even desirable at the moment. To this aim, great attention is being focused on current state-of-the-art of infrastructures to collect, organize, and share the data. Computing system architectures are undergoing rapid growth due to the establishment of cloud, virtualization, and orchestration technologies, which allows extremely complex, resilient, redundant, distributed, and almost infinitely scalable setups. At the same time, modern DBMS (Database management System) allow for heterogeneous big data, better metadata and, above all, extremely simple scalability.

The explosion of the so-called Internet of Things, allows for extended nomadic sensor networks and pervasive computing, providing fine grained, cost-effective, real-time data acquisition that represents a real boost in metadata enrichment and augmentation. All these factors, together with the growing trends in artificial intelligence (machine learning and deep learning, mostly) to build knowledge from the data, represent an intriguing challenging scenario to be exploited to disentangle the intrinsic complexity of big biological collections.

Apart from collecting, storing, and quality checking all the achievable useful data resources, the high heterogeneity of the data pushes the need for a systematic design of statistical, mathematical, computational, and bioinformatic tools aimed at analyzing and integrating them while exploiting multidisciplinary competences. Moreover, to let these data be comparable across time and space, shared protocols, reliable references, computing pipelines, and standard metadata must be established and agreed by the involved scientific communities [163], giving place to reliable long standing coordinated efforts. 

Exploiting this spatial-temporal information to build new in silico models and predictors is of great importance to widen the knowledge on marine organisms and on their biotechnological relevance. A necessary step towards this direction is also ensuring the availability and accessibility of the information generated by the scientific community, by adopting the vision of data fairness [283]. The challenge of omics data integration is pivotal to understanding biological systems, i.e., transcriptomic and proteomic data can help improve the resolution of genome annotation [284], while coupling meta-metabolomics to metagenomics will link bioactive compounds and their producers [285]. Another interesting perspective is the meta-analytical approach [286]. Plenty of data have been produced from marine resources, especially genomic and metagenomic datasets. Such data are usually analyzed to profile taxonomy and provide an overview on the main detectable functionalities. However, no platforms exist to provide standardized data processing for marine biology, such as the curatedMetagenomicData for the human metagenome [287]. 

A critical comparison and appropriate integration of results across different efforts will be an added value to identify real trends and sources of biases in the evolving area of marine biotechnology research, further leading scientific discovery forward.

## Figures and Tables

**Figure 1 marinedrugs-17-00576-f001:**
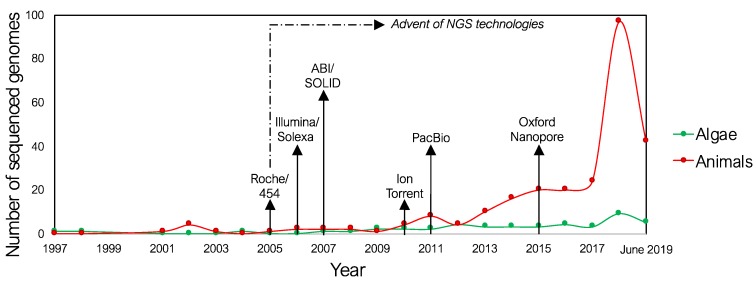
Number of sequenced genomes per year since 1997 until June 2019. Sequenced marine algae (green) and animals (red) genomes are shown. The years of the advent of next-generation sequencing (NGS) technologies as well as of the launches of the principal platforms on the market are indicated.

**Table 1 marinedrugs-17-00576-t001:** General or marine-specific reference resources/repositories per section, listed in alphabetical order.

Name	Section	Website
	**Scientific literature**	
MarinLit	Marine natural products literature	http://pubs.rsc.org/marinlit/

	**Genomics and Transcriptomics**	
AmiGO	GO functional annotation repository and analyses services	http://amigo.geneontology.org/amigo
Aniseed	Genome browser and multi-omics repository for Ascidiacea	https://www.aniseed.cnrs.fr/aniseed/
ArrayExpress	Next-generation-sequencing (NGS) data repository	https://www.ebi.ac.uk/arrayexpress/
BLAST	Local alignment versus sequence database service	https://blast.ncbi.nlm.nih.gov/Blast.cgi
CCTop	CRISPR/Cas9 target prediction tool	https://crispr.cos.uni-heidelberg.de/
CHOPCHOP	CRISPR/Cas9 and TALEN target Prediction Tool	http://chopchop.cbu.uib.no/
dbEST	Expressed sequence tag (EST) sequence repository	https://www.ncbi.nlm.nih.gov/nucleotide/
DDBJ	General multi-omics repository and analyses services	https://www.ddbj.nig.ac.jp/index-e.html
DRA	General NGS data repository	https://www.ddbj.nig.ac.jp/dra/index-e.html
Echinobase	Genome browser and multi-omics repository for Echinoderms	http://www.echinobase.org/Echinobase/
Ensembl	General multi-omics repository and analyses services	https://www.ensembl.org/
Gene Ontology	GO functional annotation repository and analyses services	http://geneontology.org/
IMG/ER	Prokaryotic sequence and function repository	https://img.jgi.doe.gov/cgi-bin/mer/main.cgi
JGI	Multi-omics repository and analyses services	https://jgi.doe.gov/
KEGG Genome	Genome sequence repository	https://www.genome.jp/kegg/genome.html
LAST	Long sequence alignment service	http://last.cbrc.jp/
Mauve	Genome alignment via homolog blocks detection	http://darlinglab.org/mauve/
MicroPan	Bacterial pangenome analysis library for R environment	https://cran.r-project.org/web/packages/micropan/index.html
NCBI	General multi-omics repository and analyses services	https://www.ncbi.nlm.nih.gov/
OIST MGU	Genome browser and analyses services for 19 marine species	https://marinegenomics.oist.jp/
PanFP	Bacterial pangenome-based functional profiles	https://github.com/srjun/PanFP
PGAWeb	Bacterial pangenome analyses service	http://pgaweb.vlcc.cn
ProGenomes	Prokaryotic sequence and functional repository	http://progenomes.embl.de/
SRA	General NGS data repository	https://www.ncbi.nlm.nih.gov/sra
	**Metagenomics and metatranscriptomics**	
dbCAN	Automated carbohydrate-active enzyme annotation	http://bcb.unl.edu/dbCAN2/
EBI Metagenomics	Microbiome sequence repository and analyses services	https://www.ebi.ac.uk/metagenomics/
Geotraces	Marine key trace elements and isotopes data repository	http://www.geotraces.org/
GLOSSary	Marine microbial sequence repository and analyses services	https://bioinfo.szn.it/glossary/
KEGG MGENES	Annotated environmental gene catalog and analyses service	https://www.genome.jp/mgenes
Marine Metagenomics Portal	Marine microbiome repository and analyses services	https://mmp.sfb.uit.no/
MG-RAST	Phylogenetic and functional analysis for metagenomics	https://www.mg-rast.org/
Ocean Gene Atlas	Analytical service for marine planktonic organisms	http://tara-oceans.mio.osupytheas.fr/ocean-gene-atlas/
Tara Oceans Database	Expedition specific raw reads sequence repository	https://www.ebi.ac.uk/services/tara-oceans-data
	**Proteomics and structural biology**	
AMBER	Molecular dynamics simulation program	http://ambermd.org/
AutoDock	Molecular docking program	http://autodock.scripps.edu/
AutoDock Vina	Multithreading program for molecular docking	http://vina.scripps.edu
CHARMM	Molecular dynamics simulations program	https://www.charmm.org/charmm/
Desmond	Molecular dynamics simulations server	https://www.schrodinger.com/desmond
DOCK	Molecular docking server	http://dock.compbio.ucsf.edu/
FlexX	Molecular docking server	https://www.biosolveit.de/FlexX/
Glide	Molecular docking server	https://www.schrodinger.com/glide
GOLD	Molecular docking program	https://www.ccdc.cam.ac.uk/solutions/csd-discovery/components/gold/
GROMACS	Molecular dynamics simulations program	http://www.gromacs.org
HHpred	Homology modelling server	https://toolkit.tuebingen.mpg.de/#/tools/hhpred
I-TASSER	Ab-initio structure prediction server	https://zhanglab.ccmb.med.umich.edu/I-TASSER/
ICM	Molecular docking program	http://www.molsoft.com/docking.html
InterPro	Protein function repository and analytical services	https://www.ebi.ac.uk/interpro/
LeDock	Molecular docking program	http://www.lephar.com/download.htm
Modeller	Homology modelling program	https://salilab.org/modeller/
MOE-Dock	Molecular docking server	https://www.chemcomp.com/index.htm
NAMD	Molecular dynamics simulations program	http://www.ks.uiuc.edu/Research/namd/
OpenMM	Molecular dynamics simulations program	http://openmm.org/
PDB	Protein structure repository	https://www.rcsb.org/
PFAM	Protein family repository	https://pfam.xfam.org/
Phyre2	Threading and ab-initio structure prediction server	http://www.sbg.bio.ic.ac.uk/~phyre2/html/page.cgi?id=index
RaptorX	Homology modelling and threading structure prediction server	http://raptorx.uchicago.edu
rDock	Molecular docking program	http://rdock.sourceforge.net/
Robetta	Homology modelling and ab-initio structure prediction server	http://www.robetta.org/
Surflex	Molecular docking program	http://www.jainlab.org/downloads.html
Swiss-model	Homology modelling server	https://swissmodel.expasy.org
SwissDock	Molecular docking server	http://www.swissdock.ch
UniProt	Protein sequence and function repository	https://www.uniprot.org/
	**Metabolomics**	
Anti-smash	Annotation and analysis of secondary metabolite biosynthesis	https://antismash.secondarymetabolites.org/#!/start
ChemSpider	Compound repository	http://www.chemspider.com/
GNPS	Tandem mass (MS/MS) spectrometry data repository	https://gnps.ucsd.edu/ProteoSAFe/static/gnps-splash.jsp
KEGG	Metabolism data repository and analyses services	https://www.genome.jp/kegg/
MEROPS	Compound repository and analyses services	https://www.ebi.ac.uk/merops/
MetaCyc	Metabolism data repository and analyses services	https://metacyc.org/
NaPDoS	Compound repository and analyses services	https://www.biokepler.org/use_cases/napdos
Reactome	Metabolism data repository and analyses services	https://reactome.org/
The Super Natural II database	Compound repository	http://bioinf-applied.charite.de/supernatural_new/index.php

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
