# Peer review of "Bioinformatics for Marine Products: An Overview of Resources, Bottlenecks, and Perspectives"

_marinedrugs, 2019, doi:10.3390/md17100576_

Round 1

Reviewer 1 Report

The rapid progress of the -omics sciences revealed novel opportunities to advance the knowledge on biological systems. In this paper, authors provide an overview of the most relevant bioinformatics resources and major approaches, highlighting perspectives and bottlenecks for an appropriate exploitation of these opportunities for biotechnology applications from marine resources. They reviewed more than two hundred papers. Their review paper is quite valuable for researchers interesting for marine products. Their paper cover many aspects for research on marine products. 

Author Response

Response to Reviewer 1 Comments

The rapid progress of the -omics sciences revealed novel opportunities to advance the knowledge on biological systems. In this paper, authors provide an overview of the most relevant bioinformatics resources and major approaches, highlighting perspectives and bottlenecks for an appropriate exploitation of these opportunities for biotechnology applications from marine resources. They reviewed more than two hundred papers. Their review paper is quite valuable for researchers interesting for marine products. Their paper covers many aspects for research on marine products. 

Reviewer 2 Report

marinedrugs-588343

Title: Bioinformatics for marine products: an overview of resources,
bottlenecks and perspectives

1. Some of the primary literature on the drug discovery from marine environment are not cited. Please cite these articles that talk about drug discovery, cyanobacteria secondary metabolites.

DOI:10.1126/science.1168243
https://doi.org/10.1517/17460441.3.8.903
https://doi.org/10.1016/j.phytochem.2007.01.012

2. On page 154, please expand the name of organism species such as C. robusta and S. purpuratus.
3. I can find IMG/M as (Integrated Microbial Genomes and Microbiomes) but what is IMG/ER ? Please expand it on its first usage.
4. Line 187, add the word “biological processes” instead of processes.
5. On page 255, was there a reference missing, I found a [...] bracket. If the reference is missing then please add that else remove these brackets.
6. On Line 268-269 you talked about pangenomic analysis of genes/gene clusters and their transfer to well characterized organism for byproducts production. I believe you are taking about heterologous production of byproducts by inserting gene/gene cluster. Few examples of such successful experiments would help the reader to understand the implication of such exercise. Please add those as this approach is quite interesting but due to lack of details as a reader I can’t relate to its importance. As I believe several commercial companies are using this approach to produce interesting chemicals at commercial scale using fermentation.
7. The paragraph from Line 270 to 301, there is lot of rambling but it don’t provide anything concrete information how to go about from sequence to a biological product. May be you can save this text by inserting a cartoon workflow how a gene for particular biological product can be identified, how it can be cloned into a heterologous host and how the product can be synthesized in large amount which ideally not possible in the native organism (perhaps because of difficulty in cultivation). Then you can write about the different considerations such as design of gene cluster, design of promoter and terminator region and how this could be transformed into a host and how a particular product can be synthesized in vivo heterologously.
8. Thank you very much for mentioning the “middle-down proteomics”. This was new to me and never know something like this exists. It is actually worth expanding it little bit more by mentioning the different enzymes used in this workflow such as Asp-N and Glu-C for preparing digest of large size peptides.
9. In the proteomics and structural biology section you talk about virtual docking using AutoDock and similar software. But these softwares are mainly used for ligand-receptor modeling and determination of binding affinity. You should also talk about newer approach such as CETSA (cellular thermal shift assay) for identification of drug target. It shows lot of promise in terms of identification of drug target from a lysate when the direct protein target of a new drug is not known.
10. Overall the review article is nicely written with comprehensive information on the resources available for marine products. Few things should be fixed in the list of references: for example, ref. No 5, a online link should be provided. Ref. No 38, name of the first author is missing, this reference need a fix.
11. Ref no 74 and 75, just add names of first 5 authors followed by et. al. that would save the space. Same for #84, 85, 123, 240, 256,

Author Response

Response to Reviewer 2 Comments 

Point 1: Some of the primary literature on the drug discovery from marine environment are not cited. Please cite these articles that talk about drug discovery, cyanobacteria secondary metabolites.

DOI:10.1126/science.1168243

https://doi.org/10.1517/17460441.3.8.903

https://doi.org/10.1016/j.phytochem.2007.01.012

Response 1: We included the suggested references:

line 51, “As other examples, the marine microbiota appears to be a promising and endless source for new drug development (Tan, 2007)” line 54, “In health sciences many marine natural products were revealed to be toxins or bioactive compounds, and were deeply studied to understand their action (Li and Vederas, 2009)” line 463, “Proteomics studies led to the discovery of peptides and toxins useful for biomedical research from sea anemone, sea sponge and cyanobacteria (Wase and Wright, 2009)”

Point 2: On line 159, please expand the name of organism species such as C. robusta and S. purpuratus.

Response 2: We expanded the names as suggested.

Point 3: I can find IMG/M as (Integrated Microbial Genomes and Microbiomes) but what is IMG/ER? Please expand it on its first usage.

Response 3: We expanded the acronyms as suggested. 

Point 4: Line 187, add the word “biological processes” instead of processes.

Response 4: We added the word as suggested.

Point 5: On line 255, was there a reference missing, I found a [...] bracket. If the reference is missing then please add that else remove these brackets.

Response 5: We used the [...] notation to cut a phrase that we literally cited from a reference. Accordingly to the comment, we removed this notation and the brackets, updating the text at line 280 as follows: “the entire genomic repertoire of a given phylogenetic clade, encoding for all possible lifestyles carried out by its organisms” 

Point 6: On Line 268-269 you talked about pangenomic analysis of genes/gene clusters and their transfer to well characterized organism for byproducts production. I believe you are taking about heterologous production of byproducts by inserting gene/gene cluster. Few examples of such successful experiments would help the reader to understand the implication of such exercise. Please add those as this approach is quite interesting but due to lack of details as a reader I can’t relate to its importance. As I believe several commercial companies are using this approach to produce interesting chemicals at commercial scale using fermentation.

Response 6: We thank the reviewer for this suggestion. We have implemented the main text with a further paragraph (line 295) showing a suitable approach in the identification of biosynthetic gene clusters through pangenomics and subsequent heterologous expression of interesting BSGCs. The added paragraph is the following: “Indeed, this approach has been applied before to identify, clone and express candidate antibiotic resistance genes in the Salinispora genus by screening more than 80 strains. This also allowed to correctly identify previously “orphan gene clusters for which function could not be assigned, by inspecting the function of their orthologs and by analysing their products through heterologous expression in suitable hosts”. We added moreover a new reference:

Tang, X.; Li, J.; Millan-Aguinaga, N.; Zhang, J. J.; O'Neill, E. C., et al., Identification of Thiotetronic Acid Antibiotic Biosynthetic Pathways by Target-directed Genome Mining. ACS chemical biology 2015, 10, (12), 2841-2849. 

Point 7: The paragraph from Line 270 to 301, there is lot of rambling but it doesn’t provide anything concrete information how to go about from sequence to a biological product. May be you can save this text by inserting a cartoon workflow how a gene for particular biological product can be identified, how it can be cloned into a heterologous host and how the product can be synthesized in large amount which ideally not possible in the native organism (perhaps because of difficulty in cultivation). Then you can write about the different considerations such as design of gene cluster, design of promoter and terminator region and how this could be transformed into a host and how a particular product can be synthesized in vivo heterologously.

Response 7: We did not mean to get into the procedure to obtain a biological product from a DNA fragment. In this paragraph we wanted to address the heterogeneity in terms of data content generated by the rapid increase of efforts in genomics and transcriptomics. Realizing that we didn’t succeed to properly address this question, we rephrased the entire paragraph (now starting at line 301) as follows: “Thanks to the diffusion of NGS technologies and the constant decrease in the sequencing costs a plethora of genomics and transcriptomics datasets have been generated. Although being associated to the same species, they often exhibit high dissimilarities in terms of data quality, curation and methodologies employed (e.g. same species with different genome annotation versions). This is due to several factors: i) data production is several order of magnitude faster than the release of exhaustively annotated and curated datasets;; ii) the opportunity to publicly release even partial and still uncomplete data, which can be anyway of interest for the scientific research, but often remain in a preliminary version; iii) the run for releasing dedicated resources for specific targets, which often causes uncoordinated parallel efforts, resulting in similar resources covering partially overlapping information; iv) the lack of rules for the withdrawal of obsolete public collections; v) the presence of software errors, such as in automatic annotation pipelines, that might generate errors not easy to be detected and that are, if not curated, inherited in subsequent versions. Navigating this overwhelming amount of resources, may cause confusion to non-expert users and lead to limited scientific applications, thus determining one of the major bottlenecks in bioinformatics. Examples of heterogeneity in terms of data content are evident even in reference platforms such as NCBI and Ensembl. As an example, for the echinoderms reference species, i.e. Strongylocentrotus purpuratus (purple sea urchin), the latest genome assembly versions in NCBI and in Ensembl are different (Spur. version 4.2 and 3.1, respectively), misleading the users and affecting the reproducibility and the comparability of the generated results outside of the used platform. Moreover, as in the case of the diatom Phaeodactylum tricornutum, although the latest available genome assembly version is the same in both NCBI and Ensembl, the genome annotation versions refer to different analytical pipelines, the NCBI Genome Annotation Pipeline  and the Ensembl Gene Annotation approach, respectively. Annotation pipelines have indeed different sensitivity in determining gene structures and in predicting CDSs, giving results that are resources dependent and do not necessarily fully overlap.” 

Point 8: Thank you very much for mentioning the “middle-down proteomics”. This was new to me and never know something like this exists. It is actually worth expanding it little bit more by mentioning the different enzymes used in this workflow such as Asp-N and Glu-C for preparing digest of large size peptides.

Response 8: We thank the reviewer for the suggestion, and changed the phrase at lines 460-462 from “Middle-down proteomics approaches, instead, generate longer peptide fragments compared to bottom-up strategies by the use of protocols involving restriction enzymes”, to “Middle-down proteomics approaches, instead, generate longer peptide fragments compared to bottom-up strategies by the use of protocols involving single-residue specific proteases such as Lys-C, Glu-C, Asp-N, and Lys-N”, adding 8 new references:

Chisolm, D. J.; Klima, J.; Gardner, W.; Kelleher, K. J., Adolescent behavioral risk screening and use of health services. Administration and Policy in Mental Health and Mental Health Services Research 2009, 36, (6), 374. Forbes, A. J.; Mazur, M. T.; Patel, H. M.; Walsh, C. T.; Kelleher, N. L., Toward efficient analysis of >70 kDa proteins with 100% sequence coverage. Proteomics 2001, 1, (8), 927-33. Wu, S. L.; Kim, J.; Hancock, W. S.; Karger, B., Extended Range Proteomic Analysis (ERPA): a new and sensitive LC-MS platform for high sequence coverage of complex proteins with extensive post-translational modifications-comprehensive analysis of beta-casein and epidermal growth factor receptor (EGFR). Journal of proteome research 2005, 4, (4), 1155-70. Sidoli, S.; Lin, S.; Karch, K. R.; Garcia, B. A., Bottom-up and middle-down proteomics have comparable accuracies in defining histone post-translational modification relative abundance and stoichiometry. Analytical chemistry 2015, 87, (6), 3129-3133. Sidoli, S.; Schwammle, V.; Ruminowicz, C.; Hansen, T. A.; Wu, X., et al., Middle-down hybrid chromatography/tandem mass spectrometry workflow for characterization of combinatorial post-translational modifications in histones. Proteomics 2014, 14, (19), 2200-11. Swaney, D. L.; Wenger, C. D.; Coon, J. J., Value of using multiple proteases for large-scale mass spectrometry-based proteomics. Journal of proteome research 2010, 9, (3), 1323-1329. Gonzalez‐Hidalgo, J. C.; Lopez‐Bustins, J. A.; Štepánek, P.; Martin‐Vide, J.; de Luis, M., Monthly precipitation trends on the Mediterranean fringe of the Iberian Peninsula during the second‐half of the twentieth century (1951–2000). International Journal of Climatology: A Journal of the Royal Meteorological Society 2009, 29, (10), 1415-1429. Taouatas, N.; Drugan, M. M.; Heck, A. J.; Mohammed, S., Straightforward ladder sequencing of peptides using a Lys-N metalloendopeptidase. Nature methods 2008, 5, (5), 405-7.

Point 9: In the proteomics and structural biology section you talk about virtual docking using AutoDock and similar software. But these softwares are mainly used for ligand-receptor modeling and determination of binding affinity. You should also talk about newer approach such as CETSA (cellular thermal shift assay) for identification of drug target. It shows lot of promise in terms of identification of drug target from a lysate when the direct protein target of a new drug is not known.

Response 9: We thank the reviewer for this suggestion, and added at the end of the molecular docking paragraph the following phrase (line 516-518): “These bioinformatics approaches, together with cutting-edge technologies able to highlight physical interaction with the target protein, such as the cellular thermal shift assay (CETSA), are essential in the design and development of new drugs.”, with a new reference.

Point 10: Overall the review article is nicely written with comprehensive information on the resources available for marine products. Few things should be fixed in the list of references: for example, ref. No 5, an online link should be provided. Ref. No 38, name of the first author is missing, this reference need a fix.

Response 10: We fixed the references as suggested.

Point 11: Ref no 74 and 75, just add names of first 5 authors followed by et. al. that would save the space. Same for #84, 85, 123, 240, 256,

Response 11: We fixed the bibliography format as suggested.

Reviewer 3 Report

The authors present a comprehensive review of the bioinformatic tools available for researchers in marine science but also for the scientific community, interested in the discovery of new products which can be used for the benefit of humans. The review is well organized and complete. I have only some comments to do.

1. Page 8, Lines 207-209: maybe authors can add some examples of the secondary metabolites and enzymes from the references in this paragraph.

2. There are some inconsistencies in formats of several words: in silico and in-silico, or omics and -omics.

3. Bottleneck section: maybe this section is too generic and I wonder if there is any specific/more serious bottleneck for marine products.

Author Response

Response to Reviewer 3 Comments 

The authors present a comprehensive review of the bioinformatic tools available for researchers in marine science but also for the scientific community, interested in the discovery of new products which can be used for the benefit of humans. The review is well organized and complete. I have only some comments to do.

Point 1: Page 8, Lines 207-209: maybe authors can add some examples of the secondary metabolites and enzymes from the references in this paragraph.

Response 1: According to this suggestion, we added the following phrase at line 228: “Some recent examples of such enzymes include the new flavin-dependent halogenase, isolated from a marine sponge metagenome, and several α-amylases isolated from a sea anemone microbial community, while metabolites range from derivatives of amino acids and nucleosides, macrolides, porphyrins, terpenoids to aliphatic cyclic peroxides and sterols”, including three new references in the text.

Point 2: There are some inconsistencies in formats of several words: in silico and in-silico, or omics and -omics.

Response 2: We fixed these inconsistencies, retaining the “in silico” and “–omics” forms.

Point 3: Bottleneck section: maybe this section is too generic and I wonder if there is any specific/more serious bottleneck for marine products.

Response 3: We thank the reviewer for the comment, however our initial decision was to discuss specific bottlenecks within each section of this review, describing the more general view before introducing the final paragraph of perspectives. To meet the comment of the reviewer, we specified this concept at line 628 with the following phrase: “Specific bottlenecks, instead, which are exclusively related to a particular area of -omics sciences, are discussed within each section of this review.”

For the sake of clarity, we report here the manuscript excerpts of the specific bottlenecks for each section:

Genomics and transcriptomics: from line 301 to line 342 (“Thanks to the diffusion of NGS technologies and the constant decrease in the sequencing costs a plethora of genomics and transcriptomics datasets have been generated. Although being associated to the same species, they often exhibit high dissimilarities in terms of data quality, curation and methodologies (e.g. same species with different genome annotation versions). This is due to several factors: i) the production of new datasets is several order of magnitude more fast than the release of exhaustively annotated and curated data collections…”) Metagenomics and metatranscriptomics: from line 382 to line 392 (“However, it is also characterized by important hindrances and potential biases, including but not limited to the lack of reference genomes (which are needed for the evaluation of potential taxonomical and genomic novelties) and standardized laboratory procedures and bioinformatics pipelines. Therefore, there is a compelling need for the development of standardized reference collections and protocols for metatranscriptomic annotation and analyses, which are still quite pioneering but might yield important results for bioprospecting.The massive flow of meta-omics sequence data highlights the need for comprehensive databases to collect the accumulating information and, additionally, appropriate curation and tools to exploit this information for taxonomical assignments and functional analyses. Until now, meta-omics-specific reference databases do not exist.”) Proteomics and structural biology: from line 526 to line 535 (“The major bottleneck of proteomics studies derives from their nature, i.e. from the complexity of biological structures and of the physiological processes in which they are involved. In particular, in structural biology applications, this complexity has an impact on the resolution of the protein structure data generated by crystallographic or NMR experiments, affecting all the downstream bioinformatics procedures described above, such as homology modelling, molecular dynamics and drug design techniques. Structures with resolution of 3 Å or higher show only the basic conformation of the protein chain, lacking any information about their atomic structure. Moreover, the need to isolate and study molecules through structure outside from their natural functional context, inhibiting their typical changes, can affect the right conformational assignments and mislead associated investigations on their behavior in biological environments.”) Metabolomics: from line 573 to line 589 (“All the presented resources rely on the correct definition of the biological function of the collected bioactive compounds and metabolites. An accurate annotation is necessary for data interpretation, however, metabolite identification is still a major bottleneck in untargeted metabolomics. Computational workflows for metabolomics interpretation, including high throughput metabolite profiling and annotation, are highly challenging tasks, with fast evolving metabolomics dataset specifically generated by dedicated service centers. Main delicate issues are due to the variability of resolution and the difficulty to establish generalized standards from different specialized laboratories and technologies. Although community guidelines for the detection of metabolites were established years ago, the adaptation to recommended standards is still far to be achieved. The complexity of metabolomics data coming from different combinations of various chromatographic and mass spectrometric acquisition methods resulted in the establishment of diverse pipelines and workflows, which often involve non-standardized manual curation. Furthermore, bioinformatics tools in the field still need to better address the problem of enrichment analyses, accurately linking metabolomics data to the most reliable similar compounds and building exhaustive pathway diagrams. These approaches need to be integrated into customizable workflows, such as the ones based on R or Python programming languages for the design of re-usable and shared software.”)

Reviewer 4 Report

The review paper entitled “Bioinformatics for marine products: an overview of resources, bottlenecks and perspectives” by Ambrosino et al. presents a thorough summary of the publicly available repositories holding all the big data that are currently accumulating through various -omics technologies, with particular emphasis on data coming from marine organisms.

I enjoyed reading this review, which presents the different currently available -omics technologies for marine compounds discovery, makes a sound compilation of different databases, and highlights potential limitations for best use of the data. I think that this review will be useful for the general reader of Marine Drugs.

I have the following suggestion that the authors may want to take into consideration when revising the manuscript:

Title, abstract, and main text: The review is focused on the different databases (and associated tools) compiling marine –omics data. Hence, the word bioinformatics is not properly used. The term Bioinformatics also includes, for instance, all the software associated to assembling and annotation genomes/ transcriptomes, which is not even mentioned. I feel that something like “Big data repositories for marine products…” will best reflect the topic of the paper. Similarly, in the abstract “an overview of most relevant data bases, resources, …” seems to me more appropriate. Line 40. Mentioning here the deep sea hot vents as example of unique ecosystem on Earth would be timely. Line 45. The adaptation to a variety of conditions… Line 47. Add also pharmacology Line 63, Please mention here explicitly WoRMS as taxonomic database devoted to marine species. Line 68. Do the authors know how many molecules from marine organisms are already approved and in the market? Line 84. Specific species might be harsh-sounding, particularly when specific is used again in the sentence. Perhaps, “concrete” or “particular” species could be a better alternative Line 97. Although mention later in the text, please clarify here that not always organisms can be isolated or cultured. Line 118. Without experimental validation, data mining alone does not discover functionally useful compounds. Hence, I suggest changing to “natural compounds discovery by pinpointing potential interesting candidates”. … Line 154. Spell out the two genera. Line 156. Please mention here (or where appropriate in the text) that the quality of the genomes is variable, and that particularly early ones based on 454 or only Illumina is not that high compare to those assembled from new technologies such a s PacBio and Nanopore, which clearly increase contiguity. Also mention that despite their lower quality, those early genomes are still good for detecting candidate new compounds but not for other comparative genomic analyses. Figure 1. Adding arrows in the years that new sequencing technologies (454, Illumina, PacBio) appeared in the market would be informative. Mention, that 2019 was obviously not complete. Not clear the rationale behind the selection of the three particular groupings. Please argue. Line 165 raw reads Line 184. Gene and gene clusters “and help annotation of genomes”. Please mention that percentages of anonymous proteins in genomes vary strongly in different taxa. Line 203. Here, the authors could use the many venom gland transcriptomes of cones as example. Line 214. Single cell transcriptomics is a very promising tool Line 235. Please mention that BLAST searches can be very shopisticated as is the case of tBLASTx that uses the six potential ORFs of a sequence. Comparing several genomes of the same taxon is also the common practice in population genomics, which carries out selection scans to detect genes under positive selection (normally associate to proteins of evolutionary relevance) Line 285. “echinoderm” Line 297. Please mention here (or where relevant in the text; e.g. lines 578-579) that automatic annotation tools produce errors, which if not curated are included in updated reference databases and are easily spread. Line 309. Could the author provide any recommendation as to how improve the situation in the future? Line 433. Cone snails Line 452. Add a reference after “structural biology” Line 502. Please highlight in this section the importance of metabolomics for studying the influence of marine organisms on climatic changes and macroecology through their effect of C-, N-, S-cycles. Line 558. I miss a mention to (the increasing lack of) taxonomic expertise as a bottleneck and the need of having vouchers in public museum collections to backup the sequence information generated. Line 558. I also miss a brief discussion on the numerous data not publicly available due to patent processes. Line 597. Could the authors provide here a year (e.g. 2010?) since which data has become more reliable? Line 606. Metadata better than data?

Author Response

Response to Reviewer 4 Comments 

Point 1: Title, abstract, and main text: The review is focused on the different databases (and associated tools) compiling marine –omics data. Hence, the word bioinformatics is not properly used. The term Bioinformatics also includes, for instance, all the software associated to assembling and annotation genomes/ transcriptomes, which is not even mentioned. I feel that something like “Big data repositories for marine products…” will best reflect the topic of the paper. Similarly, in the abstract “an overview of most relevant data bases, resources, …” seems to me more appropriate.

Response 1: We thank the reviewer for the comment. However, the scope of our paper is focused on describing the bioinformatics resources that support marine natural products exploitation, neglecting bioinformatics software, such as assembly and genome annotation programs, which are usually worthy to be mentioned in general overviews. Moreover, pointers to the relevant literature on those topics were also provided in this review (see refs. 76-77, 105-108). Overall, we presented (see Table 1 in the manuscript for details) not only 35 databases, but also 14 on-line tools and 11 software, all rather relevant for the exploitation of  biotechnology applications from marine resources. However, to better clarify the scope of the review, we added the following sentence at line 153: “Discussion of general topics not specific to marine biology, such as genomic or transcriptomic assembly and annotation, is beyond the scope of this paper. However, interested readers can find pointers to relevant literature in the related sections.” 

Point 2: Line 40. Mentioning here the deep sea hot vents as example of unique ecosystem on Earth would be timely.

Response 2: We thank the reviewer for the suggestion, and modified the text adding the following sentence at line 40: “…and adapting to a wide range of conditions – from the extreme cold of polar seas to the extremely high temperatures and pressures of deep-sea hydrothermal vents” 

Point 3: Line 45. The adaptation to a variety of conditions…

Response 3: We corrected as suggested. 

Point 4: Line 47. Add also pharmacology

Response 4: We corrected as suggested.

Point 5: Line 63, Please mention here explicitly WoRMS as taxonomic database devoted to marine species.

Response 5: We thank the reviewer for the suggestion, and updated the sentence at line 63, from “Nevertheless, the marine habitat is still poorly explored. It is estimated that, in spite of 250 years of taxonomic classification and over 1.2 million species already catalogued in reference databases, 91% of species in the ocean still awaits for a description” to “Nevertheless, the marine habitat is still poorly explored. It is estimated that, in spite of 250 years of taxonomic classification and over 1.2 million species already catalogued in reference databases such as the World Register of Marine Species, 91% of species in the ocean still awaits for a description”. Moreover, we added a new reference. 

Point 6: Line 68. Do the authors know how many molecules from marine organisms are already approved and in the market? 

Response 6: We updated the manuscript at line 77 adding the following sentence: “However, the number of approved and marketed marine natural products is still very limited (11 approved drugs, 5 of which with anti-cancer activity, and more than 20 other natural products in clinical phase, as of 2018).”. Moreover, we added a new reference.

Point 7: Line 84. Specific species might be harsh-sounding, particularly when specific is used again in the sentence. Perhaps, “concrete” or “particular” species could be a better alternative

Response 7: We thank the reviewer for the suggestion, and changed the text at line 88 from “specific species or specific ecosystems” to “specific species or particular ecosystems”

Point 8: Line 97. Although mention later in the text, please clarify here that not always organisms can be isolated or cultured.

Response 8: We thank the reviewer for the suggestion, and added at line 110 the following sentence: “On the other hand, only a fraction of the marine diversity can be aptly cultured in a laboratory; this approach must therefore be complemented by alternative techniques, in order to explore a larger portion of this diversity. In particular, molecular techniques and advanced sequencing approaches were indeed utilized to this extent to capture the genetic and genomic diversity of the unculturable fraction of marine biological diversity (especially regarding prokaryotes).”

Point 9: Line 118. Without experimental validation, data mining alone does not discover functionally useful compounds. Hence, I suggest changing to “natural compounds discovery by pinpointing potential interesting candidates”.

Response 9: We thank the reviewer for the suggestion, and updated the sentence at line 125 as follows: “The capability of performing genome and gene data mining as an essential complementary approach to traditional experimental methods greatly speeded up the process of natural compounds discovery by pinpointing potential interesting candidates”.

Point 10: Line 154. Spell out the two genera.

Response 10: We corrected (line 165) as suggested.

Point 11: Line 156. Please mention here (or where appropriate in the text) that the quality of the genomes is variable, and that particularly early ones based on 454 or only Illumina is not that high compare to those assembled from new technologies such a s PacBio and Nanopore, which clearly increase contiguity. Also mention that despite their lower quality, those early genomes are still good for detecting candidate new compounds but not for other comparative genomic analyses.

Response 11: We thank the reviewer for the suggestion, and added this sentence to the manuscript (line 167): “Resequencing efforts were in some cases necessary to improve the low quality genomes, useful to detect candidate new compounds but not for comparative genomic analyses, obtained with older or inadequate technologies”

Point 12: Figure 1. Adding arrows in the years that new sequencing technologies (454, Illumina, PacBio) appeared in the market would be informative. Mention, that 2019 was obviously not complete. Not clear the rationale behind the selection of the three particular groupings. Please argue.

Response 12: We thank the reviewer for the suggestion, and added this information in the modified figure. We also clarify that this timeline is valid till June 2019. Moreover, we decided to separate vegetal from animal sequenced genomes, reporting only the marine species for both categories.

Point 13: Line 165 raw reads

Response 13: We corrected as suggested.

Point 14: Line 184. Gene and gene clusters “and help annotation of genomes”. Please mention that percentages of anonymous proteins in genomes vary strongly in different taxa.

Response 14: We thank the reviewer for the suggestion, and changed the phrase at line 198 from “” to “In order to annotate genomes, several resources are available to provide functional information about gene and gene clusters, beyond the gene association to pathways as provided by KEGG”. Moreover, we added at the end of the paragraph (line 208) the following phrase: “However, even using different tools to functional annotate genomes, many genes still remain undefined. The percentage of anonymous genes can be very different among different species or taxa.”

Point 15: Line 203. Here, the authors could use the many venom gland transcriptomes of cones as example.

Response 15: We thank the reviewer for the suggestion, and updated the references to this sentence adding three more works about the transcriptomes of cone snails.

Point 16: Line 214. Single cell transcriptomics is a very promising tool

Response 16: We thank the reviewer for the suggestion, and changed the sentence at line 234 from: “we also mention some of the alternative NGS based sequencing approaches that are spreading to better support these investigations such as small RNA or epigenome sequencing” to “we also mention some of the alternative NGS based sequencing approaches that are spreading to better support these investigations such as small RNA, epigenome or single cell sequencing”, adding a new reference:

Ramsköld, D.; Luo, S.; Wang, Y.-C.; Li, R.; Deng, Q., et al., Full-Length mRNA-Seq from single cell levels of RNA and individual circulating tumor cells. Nature biotechnology 2012, 30, (8), 777-782

Point 17: Line 235. Please mention that BLAST searches can be very sophisticated as is the case of tBLASTx that uses the six potential ORFs of a sequence. Comparing several genomes of the same taxon is also the common practice in population genomics, which carries out selection scans to detect genes under positive selection (normally associate to proteins of evolutionary relevance)

Response 17: We thank the reviewer for the suggestion, and added at line 259 the following sentence: “Since some particular BLAST searches can be very sophisticated and computationally intensive, such as in the case of tBLASTx analyses of entire transcriptome collections that consider all the six potential ORFs of each sequence, parallel efforts spread. Complete genome alignments can be carried out…”

Point 18: Line 285. “echinoderm”

Response 18: We corrected as suggested.

Point 19: Line 297. Please mention here (or where relevant in the text; e.g. lines 578-579) that automatic annotation tools produce errors, which if not curated are included in updated reference databases and are easily spread.

Response 19: We thank the reviewer for the suggestion, and added the sentence at line 311: “v) the presence of software errors, such as in automatic annotation pipelines, that might generate errors not easy to detect and that are, if not curated, spread in the subsequent versions.”

Point 20: Line 309. Could the author provide any recommendation as to how improve the situation in the future?

Response 20: We thank the reviewer for the suggestion, and added the following sentences at line 335: “To mitigate this issue, the straightforward direction should be that reference websites (e.g. NBCI, Ensembl) should share, cross-reference and integrate more infomation, even coming from smaller consortia efforts, and clearly report updates and errors, if any. A complementary approach could be provided by experts in the field that could produce smaller but effective resources by releasing manually curated information on selected species. As an example, the GENOMA platform, an ongoing project, at the moment collects and integrates genomic information about four different marine species, reporting statistics and comparisons among several genomics resources, also through dedicated genome browsers.” We added, moreover, a reference for the GENOMA platform.

Point 21: Line 433. Cone snails

Response 21: We thank the reviewer for the suggestion, and added “cone snails” to the text (line 463) as follows: “Proteomics studies led to the discovery of peptides and toxins useful for biomedical research from sea anemone, sea sponge, cone snails and cyanobacteria”. We added, moreover, a new reference.

Point 22: Line 452. Add a reference after “structural biology”

Response 22: We thank the reviewer for the suggestion, and added the following reference: “Congreve, M.; Murray, C. W.; Blundell, T. L., Structural biology and drug discovery. Drug discovery today 2005, 10, (13), 895-907.”

Point 23: Line 502. Please highlight in this section the importance of metabolomics for studying the influence of marine organisms on climatic changes and macroecology through their effect of C-, N-, S-cycles.

Response 23: We thank the reviewer for the suggestion, and added the following phrase at line 545: “In this context, metabolomics helps the evaluation of the impact of climate changes on marine organisms, unraveling contributions that marine systems could play in mitigating the effects of global warming”. Moreover, we added two new references.

Point 24: Line 558. I miss a mention to (the increasing lack of) taxonomic expertise as a bottleneck and the need of having vouchers in public museum collections to backup the sequence information generated.

Response 24: We thank the reviewer for the suggestion, and added a brief paragraph at line 614 detailing those issues as follows: “On the other hand, prior to any bioinformatic approach, a correct taxonomic information on the specimens used is paramount for the production of high-quality research: sadly, the number of expert taxonomists in many fields of biologic research is dwindling down, in some cases due to the difficulties taxonomists face in the identification of specimens. In addition, specialized backup facilities are also required to maintain voucher collections for future inquiries.”

Point 25: Line 558. I also miss a brief discussion on the numerous data not publicly available due to patent processes.

Response 25: We thank the reviewer for the suggestion, and added a brief paragraph at line 595: “It is evident from the many examples given above that marine organisms represent an ever increasing subject of scientific investigation. Of course, they result also of great commercial interest for big (chemical, pharmacological, biotech) companies, especially for those species belonging to the so-called marine areas beyond national jurisdiction (ABNJ), that cover the 64 percent of the surface of the oceans and nearly 95 percent of its volume. Usually rich entities, such as big companies or first-world universities, which invested money in the collection and analysis of marine organisms, secure their findings by national or international patents, thus denying the scientific community a free access to their results. Even though the Nagoya protocol (2010) has somehow addressed the need of regulation in the field, the question on who is the owner of the Ocean's biodiversity data still remains open, and what kind of legal and political action are needed to prevent an unfair appropriation of marine data is still subject of debate”. We added, in addition, a new reference:

Blasiak, R.; Jouffray, J.-B.; Wabnitz, C. C. C.; Sundström, E.; Österblom, H., Corporate control and global governance of marine genetic resources. Science advances 2018, 4, (6), eaar5237

Point 26: Line 597. Could the authors provide here a year (e.g. 2010?) since which data has become more reliable?

Response 26: In order to meet this comment, we provided a time frame to the mentioned concept changing the phrase at line 648 from “If in the past the budget requirements to store data was quite disadvantageous, nowadays cheaper technologies make a catch-all approach possible or even desirable” to the following one: “If in the past the budget requirements to store data was quite disadvantageous, during the last decade storage costs became cheaper, thanks to hardware technologies and cloud commercial offers, making a catch-all approach possible or even desirable at the moment.” 

Point 27: Line 606. Metadata better than data?

Response 27: We modified as suggested.